# From Immunosenescence to Aging Types—Establishing Reference Intervals for Immune Age Biomarkers by Centile Estimation

**DOI:** 10.3390/ijms241713186

**Published:** 2023-08-24

**Authors:** Peter Bröde, Maren Claus, Patrick D. Gajewski, Stephan Getzmann, Edmund Wascher, Carsten Watzl

**Affiliations:** Leibniz Research Centre for Working Environment and Human Factors (IfADo), Ardeystraße 67, 44139 Dortmund, Germany; claus@ifado.de (M.C.); gajewski@ifado.de (P.D.G.); getzmann@ifado.de (S.G.); wascher@ifado.de (E.W.); watzl@ifado.de (C.W.)

**Keywords:** immunosenescence, biological age, biomarker, flow cytometry, longitudinal study

## Abstract

Immunological aging type definition requires establishing reference intervals from the distribution of immunosenescence biomarkers conditional on age. For 1605 individuals (18–97 years), we determined the comprehensive immune age index IMMAX from flow-cytometry-based blood cell sub-populations and identified age-specific centiles by fitting generalized additive models for location, scale, and shape. The centiles were uncorrelated with age and facilitated the categorization of individuals as immunologically slow or fast aging types. Using its 50th percentile as a reference, we rescaled the IMMAX to equivalent years of life (EYOL) and computed the immunological age gap as the difference between EYOL and chronological age. Applied to preliminary baseline and follow-up measurements from 53 participants of the Dortmund Vital Study (Clinical-Trials.gov Identifier: NCT05155397), the averaged changes in the IMMAX and EYOL conformed to the 5-year follow-up period, whereas no significant changes occurred concerning IMMAX centiles and age gap. This suggested that the participants immunologically adapted to aging and kept their relative positions within the cohort. Sex was non-significant. Methodical comparisons indicated that future confirmatory analyses with the completed follow-up examinations could rely on percentile curves estimated by simple linear quantile regression, while the selection of the immunosenescence biomarker will greatly influence the outcome, with IMMAX representing the preferable choice.

## 1. Introduction

Molecular biomarkers have their role alongside other markers in characterizing biological age [1] and in particular the aging immune system [2]. The latter process, termed immunosenescence, describes the age-related deterioration of the immune system, which does not necessarily parallel chronological age [3]. Several simple markers, based on cell type frequencies of, e.g., natural killer (NK) and T cells, total and memory/naïve sub-populations of CD4pos and CD8pos T cells, or CD8pos CD28neg T cells, were proposed as immunosenescence biomarkers [4,5,6,7,8,9,10], while composite scores combine several markers for a comprehensive evaluation of immunological aging [11,12,13]. Among those, the immune age metric IMM-AGE [11] is considered one of the most advanced immunosenescence biomarkers [14,15,16,17,18]. IMM-AGE was developed from individual longitudinal profiles of composite multi-omics data on blood cell phenotypes, functional tests with stimulated cells, and gene-expression analyses. In clinical and non-clinical settings, it has demonstrated its predictive capacity concerning cardiovascular disease and mortality [11], the risk of sepsis in trauma patients [19], the age-related decline in cardiorespiratory fitness [20] and work ability [21], and the efficacy of vaccination against SARS-CoV-2 [22]. Notably, all these application studies had to approximate the original IMM-AGE metric by a compatible set of gene expression or blood cell type data. Recently, we established an approximation termed IMMAX (immune age index) based on a small set of blood cell frequencies measured by flow cytometry, which enabled the estimation of IMM-AGE with reasonable accuracy in application studies [20,21].

Aging biomarkers, also termed aging clocks [1], are usually constructed in relation to or as predictors of chronological age, employing a variety of methods like simple linear regression, multivariate statistical models, or machine learning algorithms [23]. For comparison to chronological age, they can be expressed as equivalent years of life (EYOL), defined as the chronological age of a reference exhibiting the same aging clock level [24]. The reference value is commonly chosen as typical for the population under consideration, e.g., as the age-specific mean or median (50th percentile), which in a simple approach could be derived by regressing the biomarker on chronological age [11]. Notably, this is analogous to the established concept of expressing multivariate indices of the thermal environment on an equivalent temperature scale [25]. Rescaling to commonly known units, i.e., years for biological or immunological aging, similarly to using temperature concerning thermal stress, will not only facilitate the communication of research output to the public: it also allows for the categorization of individuals’ aging types (ageotypes) by defining age-specific reference intervals [26], and for the easy calculation of the age gap as the difference in years between the biological and chronological age [1]. Thus, when calculated with a biomarker of immune age, the age gap will represent the rate of immunological aging compared to a population reference, where positive age gap values indicate fast or accelerated aging and negative values mark a slow or decelerated aging of an individual’s immune system independently from chronological age [1]. This could also ameliorate methodical issues regarding the multicollinearity induced by the correlation of immunosenescence biomarkers with chronological age in the analysis and design of aging studies [21]. However, corresponding data and schemes for deriving aging types from immunological biomarkers are lacking, specifically concerning the comprehensive metric IMMAX.

Therefore, in analogy to pediatric growth standards [27], we established age-adjusted reference centiles covering the adult age range for the immunosenescence biomarker IMMAX by fitting generalized additive models for location, scale, and shape (GAMLSS) [28] and quantile regression models [29,30] to 1605 observations. The data were pooled from the original IMM-AGE study [11], the Dortmund Vital Study (DVS) [20,31], and a study on the efficacy of vaccination against SARS-CoV-2 (VAC) [22]. With these IMMAX centiles, which were uncorrelated to chronological age by definition, we could define different levels of slow and fast aging types, rescale IMMAX as equivalent years of life (EYOL), and calculate immunological age gap values. While EYOL represents a rescaled version of the IMMAX biomarker, the age gap could serve as a replacement for the IMMAX centiles expressed on a years-of-life scale, as both variables correlated almost perfectly and were independent of chronological age. When applied to preliminary longitudinal data from the ongoing Dortmund Vital Study (ClinicalTrials.gov Identifier: NCT05155397) [31], we observed changes from the baseline examinations after a 5-year follow-up in IMMAX and EYOL, which were consistent in magnitude with the follow-up period length. The non-significant changes in the IMMAX centiles and the age gaps suggest that the participants kept their position within the cohort during follow-up.

Although our preliminary findings will require confirmatory analyses with the completed follow-up examinations, the immunosenescence biomarker IMMAX together with the derived metrics EYOL and age gap showed the potential to satisfy the desiderata for clocks representing the rate of immunological aging in longitudinal research [1].

## 2. Results

As recently reported [20], the comprehensive immune age metric IMM-AGE was reasonably well approximated by IMMAX (Figure 1a). Although the age distributions were very heterogeneous between the three studies, the age-dependent increase in IMMAX was quite similar (Figure 1b), with our data filling the blank interval in the bimodal age distribution of the original IMM-AGE study [11], which had enrolled two distinct groups of young and very old adults. This justified pooling the three datasets for the subsequent centile estimation.

### 2.1. IMMAX Centile Estimation, Immunological Aging Types, and Age Gap

Figure 1c visualizes the IMMAX distributions estimated by GAMLSS for selected values of chronological age. They did not only vary by location, as indicated by the curvilinear increase in the medians with age but also by shape and spread. Whereas the IMMAX centiles correlated with the biomarker values via the age-dependent cumulative distribution functions (Figure 1d, right panel), they were uncorrelated with age by construction (Figure 1d, left panel). The resulting percentile curves in Figure 1e, with centiles chosen at intervals approximately 1 SD apart for a Gaussian distribution [27], provided reference intervals defining ordered categories of fast (or accelerated) and slow (or decelerated) immunological aging, respectively. Immune age on an equivalent-years-of-life (EYOL) scale for given values of age and immune age biomarker IMMAX could then be defined as the chronological age yielding the identical IMMAX value on the reference curve, chosen as P50 (Figure 1e). The immunological age gap was then calculated as the difference between immune age and chronological age. Scatterplots in Figure 1f showed a perfect correlation between IMMAX and EYOL following the non-linear course of the reference curve P50, which had to be extrapolated to very high and even negative age for mapping the whole zero-to-one range of IMMAX to EYOL values (Figure A1 in Appendix A). IMMAX centiles almost perfectly correlated with the age gap, while the correlation of the age gap with IMMAX mirrored the relation of IMMAX with the IMMAX centiles from Figure 1d. Thus, while EYOL is a re-scaled version of the biomarker IMMAX, the age gap almost perfectly substituted the IMMAX centiles, with negative values representing slow aging and positive age gaps indicating fast or accelerated immunological aging, respectively. Both the centiles and the age gap were uncorrelated with chronological age (Figure 1d), confirming the independence of this characterization of immunological aging from chronological age.

### 2.2. Sensitivity to Centile Estimation Modeling Strategy

Individual centile values are not necessarily required for the determination of aging type and age gap, which rely on pre-specified percentile curves only (Figure 1e). Such curves defining reference intervals can be fitted by different modeling strategies in addition to GAMLSS, e.g., by linear quantile regression (LQR) [29] or non-parametric quantile regression (NQR) employing regression splines [30]. Figure 2a compares the resulting percentile curves for the three algorithms, which showed quite a similar trend in the low to median age range. At old age, the GAMLSS curves for high and low percentiles wriggled compared to the smooth curves for LQR and NQR, probably due to the higher sensitivity of the GAMLSS algorithm to data sparsity at high age. These differences only had negligible consequences on aging type classification, as shown by Figure 2b, indicating that the different models classified the overwhelming proportion of individuals to identical aging types, with misclassifications only occurring between neighbored categories. Figure 2c quantifies this excellent agreement by values of Krippendorff’s α [32] above 0.9 for all comparisons. The corresponding results for the test data in Figure 2b,c corroborated the outcome for the more comprehensive training data.

The high agreement between the three modeling strategies for estimating percentile curves was confirmed when repeating the analyses for the five immune age biomarkers used for the calculation of IMMAX (Figure 3). On the other hand, concerning inter-biomarker agreement, Krippendorff’s α comparing aging type classification by the six biomarkers dropped to values below 0.2 independently from the chosen algorithm (Figure 3c, right panel), towards zero, which marks the expected coincidence of the ratings due to chance [32]. Bivariate correlation analysis (Figure 3b) revealed considerable heterogeneity between the rating algorithms, with ratings based on the memory-to-naïve ratio of CD8 cells similar to those using IMMAX, whereas close to zero, or even negative correlations with IMMAX-based ratings occurred for the NK:T cell and CD4:CD8-ratios, respectively.

Similar to aging type rating, the alternative models could also be applied for defining reference curves, usually taken as the median (P50), when calculating EYOL and age gap. Figure A1 presents the resulting curves for linear quantile regression (LQR) in addition to GAMLSS showing that with both approaches, the P50 curve needed extrapolation beyond the limits of adult age for determining EYOL over the whole zero-to-one range of the immune age biomarker IMMAX. For NQR, the extrapolation was not feasible due to the class of spline functions utilized by this model [30]. In search of an algorithm keeping EYOL values in the adult age range, we defined a hypothetical reference curve based on the IMMAX centiles calculated by GAMLSS (GAMLSS-HYP). Here, we assumed a percentile profile progressively increasing with age starting with the 0.5% percentile at the age of 19 years rising to the 99.5% percentile at 99 years by 1.24% per year, with the minimum value (0) assigned to 18 years and the maximum (1) to 100 years of age (Figure A1). The resulting age gaps based on P50 from GAMLSS and LQR were almost perfectly correlated and exhibited a coinciding switch from negative to positive values (Figure 4a), while correlations for GAMLSS-HYP were lowered, accompanied by a blurred transition from negative to positive age gaps (Figure 4a). In addition, the progressively increasing profile of reference percentiles in GAMLSS-HYP reduced the range of the corresponding age gap values compared to GAMLSS-P50 and LQR-P50 but induced a negative correlation with age (Figure 4b). On the other hand, similar to the IMMAX centiles (Figure 1d), age gaps based on GAMLSS-P50 and LQR-P50 were uncorrelated with chronological age (Figure 4b). In any case, the outcomes for the test data matched and supported the results from the training data.

### 2.3. Application to Longitudinal Data from the Dortmund Vital Study

We applied our modeling to preliminary data from 53 participants (28 females, Table A2 in Appendix A) of the ongoing longitudinal Dortmund Vital Study (DVS, ClinicalTrials.gov Identifier: NCT05155397) [31], who had completed the baseline examinations and the first follow-up five years later. Longitudinal effects were expressed by the changes from baseline (Δ) in IMMAX, IMMAX centiles, EYOL, and age gap, as summarized in Table A2. Notably, for a given follow-up period length (FPL in years), the changes from baseline in EYOL and age gap are algebraically linked by ΔEYOL = ΔAge gap + FPL.

IMMAX increased significantly at the 5-year follow-up, whereas no significant change was observed for the centiles (Figure 5a, Table A2). These outcomes were mirrored on the years-of-life scale (Figure 5b) by significantly positive ΔEYOL and no significant change in the age gap. For the 50th percentile reference curves (GAMLSS-P50, LQR-P50), the values of ΔEYOL conformed to the 5-year follow-up period (Table A2). Although the corresponding changes in the hypothetical reference (GAMLSS-HYP) were lower compared to the other reference curves, these differences were not statistically significant (Figure 5b). In addition to the algebraic identity between ΔEYOL and ΔAge gap, changes from baseline (Δ) in IMMAX, IMMAX centiles, EYOL, and age gap were highly correlated (Figure 5c). No significant sex effects could be established, neither concerning the correlation (Figure 5c) nor between the distributions of the longitudinal changes, which might be related to the sample size limitations of these preliminary data (Figure 5d).

## 3. Discussion

Aging clock design including immunosenescence biomarkers should preferably involve longitudinal measurements [1]. In addition, it should allow for the categorization of the rate of aging of the individual immune system by defining age-specific reference intervals [24,26]. Of particular interest in this regard will be the comparison of individual clock levels to a defined reference population mapping the biomarker to a years-of-life scale for the easy calculation of the age gap as the difference in years between the biological and chronological age [1]. The immunosenescence biomarker IMMAX [20] will likely comply with the first requirement because it approximates the IMM-AGE score, constructed from trajectories capturing the individual’s immune-aging process [11]. Satisfying the additional desiderata concerning IMMAX as a composite immune age metric was the major objective of this study.

Our aims were accomplished by adopting concepts from pediatric growth-chart modeling [27]. The age-adjusted centile estimation for the pooled molecular immunosenescence biomarker data enabled the characterization of the comprehensive immune age metric IMMAX in terms of fast or slow aging types covering the adult age range from 18 to 97 years. As the IMMAX centiles are uncorrelated with age by construction, they do not constitute an immune age biomarker, which has to correlate with age by definition [1,23]. Instead, the centiles represent the rate of aging of the individual immune system compared to a population reference independent of age [1], which helps mitigate issues concerning multicollinearity in the analysis and design of prospective studies involving both immunosenescence biomarkers and chronological age [21]. In addition, the comparison of the biomarker values to a reference curve, commonly chosen as median curve P50 [1,11,23], facilitated the rescaling of IMMAX as equivalent years of life EYOL and calculating the immunological age gap [1]. While EYOL is a rescaled version of the biomarker IMMAX, the age gap did not correlate with age, as did the IMMAX centiles, which in turn were almost perfectly correlated with the age gap. Thus, EYOL and age gap, both employing age as a commonly known scale, may help communicate study outcomes concerning the biomarker IMMAX and the IMMAX centiles, respectively, to other researchers or to the public and policymakers. Notably, this is consistent with the established concept of expressing multivariate indices of the thermal environment on an equivalent temperature scale [25].

The preliminary longitudinal data from the DVS featured the complemental use of these metrics with the observed change in IMMAX corresponding to the 5-year follow-up period when expressed as EYOL, facilitating the interpretation that, immunologically, the sample aged as expected given the length of the follow-up period. On the other hand, the non-significant changes in the values of the IMMAX centiles and age gap indicate that, with respect to immunological aging, the individuals kept their relative positions in the cohort. In summary, this might suggest the interpretation of the observed changes as an adaptation to the aging process during the follow-up period, rather than as dysregulation of the immune system, which might be associated with individual excessive alterations contributing to the observed inter-individual variability [33,34].

Although females exhibited lower age-related levels of immunosenescence biomarkers, including IMMAX, in previous cross-sectional studies [11,20], we could not demonstrate any significant sex-related longitudinal difference. However, this might be due to the limited sample size, and thus, these preliminary findings will require confirmatory analyses through follow-up examinations. Such studies might also aim at explaining the considerable inter-individual variability in immunological aging in relation to the personal, behavioral, environmental, and work-related data gathered in the DVS [31]. The methodological comparisons performed in our study suggest that such future analyses could rely on the simpler approach involving linear quantile regression (LQR), while the selection of the immune age biomarker will require careful consideration, where the composite IMMAX metric is deemed preferable to single molecular immunosenescence biomarkers.

In conclusion, we believe that the immune age index IMMAX will have the potential to satisfy the desiderata for immunological aging clocks in longitudinal research [1,24]. In combination with the metrics EYOL and age gap established from our IMMAX data, it will facilitate communicating research findings involving the aging immune system on a familiar years-of-life scale to a broad audience including stakeholders in the scientific community, patient groups, and policymakers.

## 4. Materials and Methods

### 4.1. Datasets

Aiming at a comprehensive database for estimating age-dependent centiles of immunosenescence biomarkers, we combined the published raw data from the IMM-AGE study (N = 434) [11] with corresponding own measurements from the baseline examinations of the ongoing longitudinal Dortmund Vital Study (DVS, N = 597) [20] and a recent study concerning the efficacy of vaccination against SARS-CoV-2 (VAC, N = 574) [22]. The pooled sample comprised 1605 participants aged 18 to 97. We did not consider sex as a modifying factor because it was not available for the IMM-AGE data. For evaluating the centile estimation models, another independent set of 125 observations, mostly from firefighters [35], was augmented by 25 observations obtained from the pooled data by age-stratified subsampling. Thus, the training data for model development comprised N = 1580 participants, while the test data sample size was 150. Table A1 in Appendix A compares the training and test data concerning the distribution of age and immunosenescence biomarkers determined by flow cytometry, as described below.

### 4.2. Molecular Biomarkers of Immunosenescence by Flow Cytometry

A set of relative blood cell frequencies related to aging and senescence (NK and T cells, total and memory/naïve sub-populations of CD4pos and CD8pos T cells, and CD8pos CD28neg T cells) has been published as supplemental data to the IMM-AGE study [11]. For both the DVS and VAC samples, we determined those parameters by flow cytometry, as detailed elsewhere [20,36]. All antibodies were individually titrated to determine the optimal dilution, as shown in Table 1. Briefly, for the DVS sample, we collected peripheral venous blood from the participants in heparinized monovettes (Sarstedt, Nümbrecht, Germany), isolated peripheral blood mononuclear cells (PBMCs) by Ficoll density gradient centrifugation (PAN-Biotech, Aidenbach, Germany), and stored the cells at −170 °C for up to 6 months. Immediately after thawing for analysis, the PBMCs were kept on ice during the entire staining procedure. We stained 0.2 × 10^6^ cells with the indicated antibody cocktails for 20 min in the dark at 4 °C and subsequently washed them with FACS buffer (PBS/2% FCS). Cells were resuspended in FACS buffer. For the VAC sample, peripheral venous blood was collected in EDTA-monovettes (Sarstedt, Nümbrecht, Germany) and 100 µL whole blood was stained directly for the markers indicated in Table 1. Subsequently, samples were subjected to erythrocyte lysis, washed, and resuspended in FACS buffer. All samples were kept on ice until analysis on the same day on a BD LSRFortessa (BD Biosciences, Franklin Lakes, NJ, USA). Figure 6 displays the gating strategy with antibody panels for analyzing the lymphocytes for the NK:T cell ratio, CD4:CD8 T cell ratio, memory:naïve sub-populations of CD4pos and CD8pos T cells, and CD28neg CD8pos T cells. Relative cell frequencies were determined using the FlowJo software version 10.8.2 (FlowJo LLC, Ashland, OR, USA).

### 4.3. Data Analysis and Statistics

As described previously [20], we approximated the comprehensive immune age metric IMM-AGE by IMMAX (immune age index), which was estimated by principal component regression utilizing as predictors the flow-cytometry-based relative blood cell frequencies (Table A1). With the training data, we estimated age-specific IMMAX centiles by fitting generalized additive models for location, scale, and shape (GAMLSS) [28,37]. From the centiles, we calculated reference curves PXX, with XX (=03, 15, 50, 85, and 97) denoting the XXth percentiles chosen at intervals roughly 1 SD apart for a Gaussian distribution, thus approximately covering the mean ± 2 SD in the range [27]. For comparison, we computed the reference percentile curves PXX applying linear quantile regression (LQR) [29,38] and nonparametric quantile regression (NQR) utilizing penalized regression splines [30,39]. The reference intervals defined by the resulting PXX curves were utilized to classify the individuals into different levels of slow (below the median curve P50) or fast (above P50) immunological aging. The agreement between the different aging type rating algorithms in relation to the modeling strategies (GAMLSS, LQR, and NQR) and the immune age biomarkers was evaluated by Spearman’s correlation coefficient (ρ), and by Krippendorff’s α [32,40]. The latter is defined as α = 1 − D_o_/D_e_, with D_o_ denoting the proportion of disagreement observed in the sample and D_e_ the corresponding proportion expected by chance assignments of aging types. Thus, α = 1 will indicate perfect agreement, while α = 0 represents agreement due to chance. Finally, we rescaled the immune age biomarker IMMAX as equivalent years of life (EYOL), defined as the chronological age, for which the reference curve, commonly chosen as P50, exhibits the identical IMMAX value. We calculated EYOL utilizing different reference populations, i.e., P50 calculated with GAMLSS and LQR, and in comparison to a hypothetical reference (HYP) characterized by the GAMLSS percentiles progressively increasing from the minimum to maximum with age. The difference between EYOL and chronological age then defines the immunological age gap, with positive values indicating fast or accelerated aging, and negative values referring to slow or decelerated aging.

### 4.4. Preliminary Longitudinal Data from the Dortmund Vital Study

We illustrate the application of our modeling approach using preliminary longitudinal data comprising the immune age IMMAX from the first 53 participants of the DVS [31], who had completed the baseline examinations and the first follow-ups five years later. Wilcoxon paired sample tests and correlation analyses were utilized to assess the change from baseline in IMMAX and IMMAX centiles, as well as in EYOL and age gap determined by different reference curves (GAMLSS-P50, LQR-P50, and GAMLSS-HYP). In addition, we compared the distribution of these quantities between the 28 female and 25 male participants by Wilcoxon tests for independent samples.

All calculations were performed using R version 4.3.1 (R Core Team, Vienna, Austria) [41]. The analysis scripts and data for reproducing the results of this study are available at the source provided below in the ‘Data Availability Statement’.

## Figures and Tables

**Figure 1 ijms-24-13186-f001:**
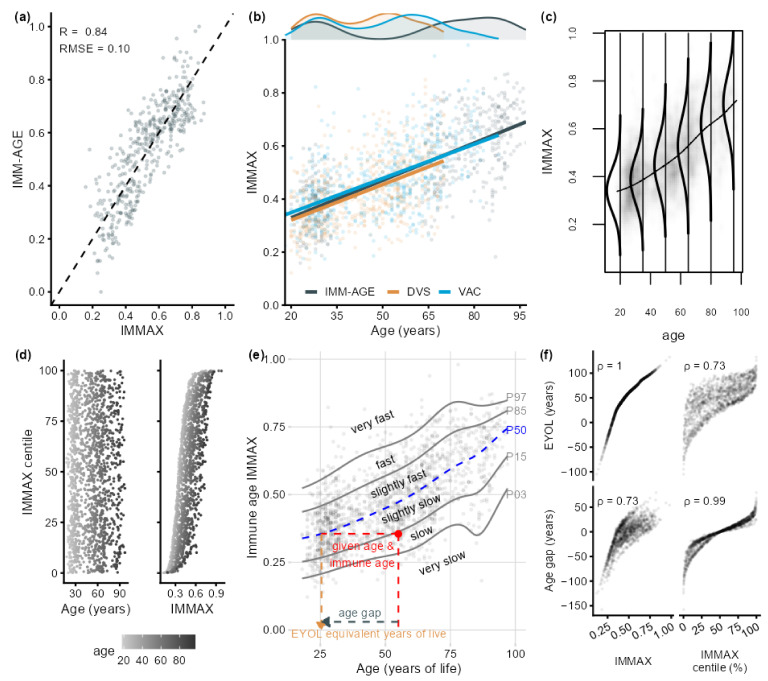
Moving from immune age biomarkers to aging type and age gap definition. (**a**) Approximation of the comprehensive IMM-AGE metric [11] by the immune age index IMMAX [20], with dashed line of identity, Pearson correlation coefficient (R), and root-mean-square prediction error (RMSE). (**b**) IMMAX related to age with regression lines in the IMM-AGE study, Dortmund Vital Study (DVS), and vaccination study (VAC), respectively, and with the age distributions from the three study groups overlaid on top. (**c**) IMMAX distribution conditional on age with the line connecting the median values estimated for the pooled training data (N = 1580) by a generalized additive model for location, scale, and shape (GAMLSS). (**d**) IMMAX centiles estimated by GAMLSS in relation to age and IMMAX, respectively. (**e**) Percentile curves (PXX, XX denoting the percentile) estimated by GAMLSS for the rating of IMMAX values into six aging type categories and for re-scaling IMMAX to equivalent years of life (EYOL) with the median curve (P50) as a reference, and the immunological age gap defined by the difference between EYOL and chronological age. (**f**) Spearman’s rank correlations (ρ) of IMMAX and IMMAX centiles with EYOL and age gap, respectively.

**Figure 2 ijms-24-13186-f002:**
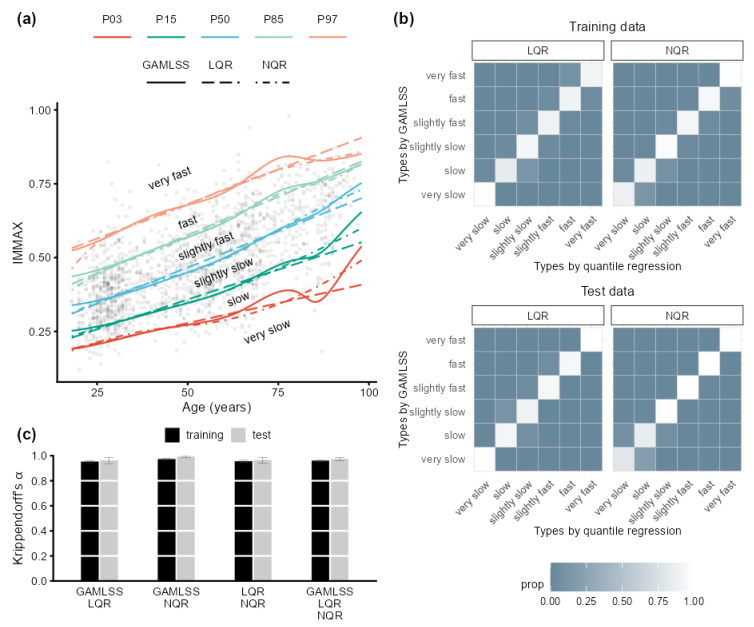
Influence of centile estimation models on the rating of immune aging types. (**a**) Age-dependent percentile curves (PXX, with XX denoting the percentile) for IMMAX with corresponding aging types estimated by GAMLSS, linear (LQR), and non-parametric quantile regression (NQR), respectively. (**b**) Confusion matrices with the proportion (prop) of immune aging type combinations rated by GAMLSS compared to the quantile regression models LQR and NQR, respectively, for the training (upper panel) and test data (lower panel). (**c**) Krippendorff’s α with error bars indicating 95% CIs for assessing the agreement between the immune aging types rated by pairwise and triple combinations of the different models for both training and test data.

**Figure 3 ijms-24-13186-f003:**
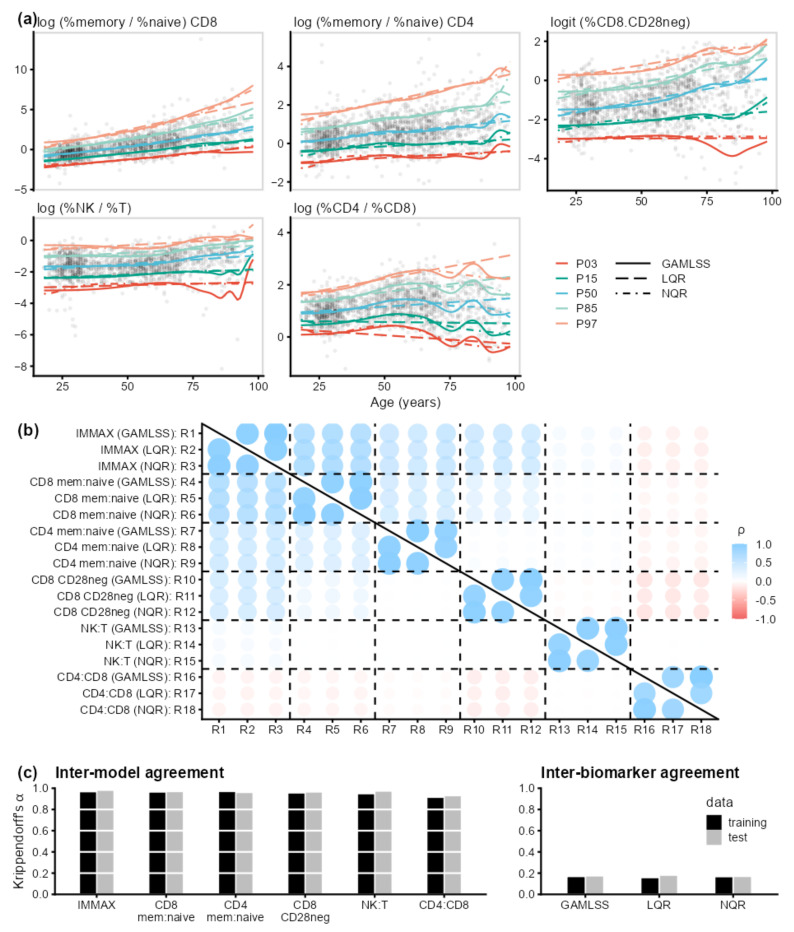
Influence of centile estimation models and choice of biomarker on immune aging types. (**a**) Age-dependent percentile curves (PXX, with XX denoting the percentile) estimated by GAMLSS, linear (LQR), and non-parametric quantile regression (NQR) for the five immunosenescence biomarkers predicting IMMAX: the cell frequency ratios of memory to naïve CD4 and CD8 T cells, respectively, of NK to T cells, of CD4pos to CD8pos T cells, and the frequency of CD28neg–CD8pos T cells. (**b**) Bivariate Spearman’s rank correlations (ρ) between the immune aging types rated by 18 algorithms R1–R18 (6 biomarkers × 3 models) for the training data (upper triangular matrix) and test data (lower triangular matrix). (**c**) Krippendorff’s α assessing the agreement between the immune aging type ratings by the different models for the six biomarkers (left panel), as well as by the different biomarkers in relation to the three models (right panel), for both training and test data.

**Figure 4 ijms-24-13186-f004:**
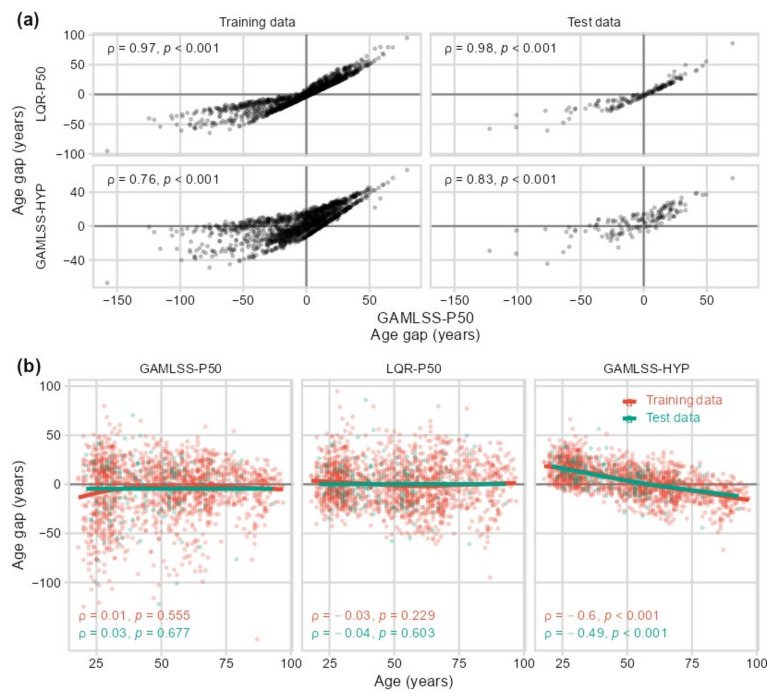
Influence of reference curves, defined as the 50th percentile (P50) estimated by GAMLSS and LQR, respectively, or assuming a hypothetical reference with percentiles gradually progressing with age (GAMLSS-HYP), on the immunological age gap. Spearman’s rank correlations (ρ) for training and test data (**a**) between age gaps defined by three reference curves, and (**b**) of age gap vs. chronological age with overlaid smoothing splines.

**Figure 5 ijms-24-13186-f005:**
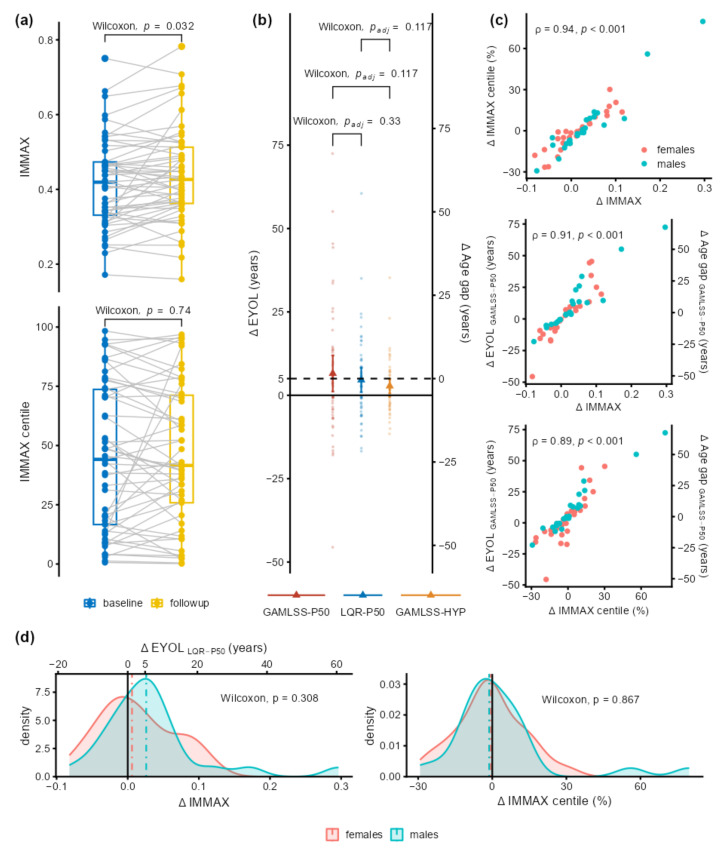
Preliminary longitudinal data from the DVS. (**a**) Baseline and follow-up values from 53 participants of IMMAX (upper panel) and IMMAX centiles (lower panel) with statistical significance assessed using the Wilcoxon paired sample test. (**b**) Change from baseline in equivalent years of life (ΔEYOL) derived from IMMAX with different reference curves defined as the 50th percentile estimated by GAMLSS and LQR, respectively, or assuming a hypothetical reference with percentiles gradually progressing with age (GAMLSS-HYP). Triangles and error bars indicate means with 95% CIs, the dashed horizontal line marks the 5-year follow-up period, corresponding to the zero-value for the change in age gap (ΔAge gap). Pairwise statistical comparison using the Wilcoxon paired sample test adjusted for multiple testing. (**c**) Pairwise scatterplots with Spearman’s correlation coefficients (ρ) for the change from baseline (Δ) in IMMAX, IMMAX centile, and EYOL calculated with the GAMLSS-P50 reference. (**d**) Density plots comparing the female and male distributions of ΔIMMAX (left panel, with upper horizontal axis showing the linearly rescaled equivalent years of life ΔEYOL_LQR-P50_), and ΔIMMAX centiles (right panel). Vertical dot–dashed lines indicate medians by sex; solid lines mark zero effects. Statistical comparisons by independent-sample Wilcoxon test.

**Figure 6 ijms-24-13186-f006:**
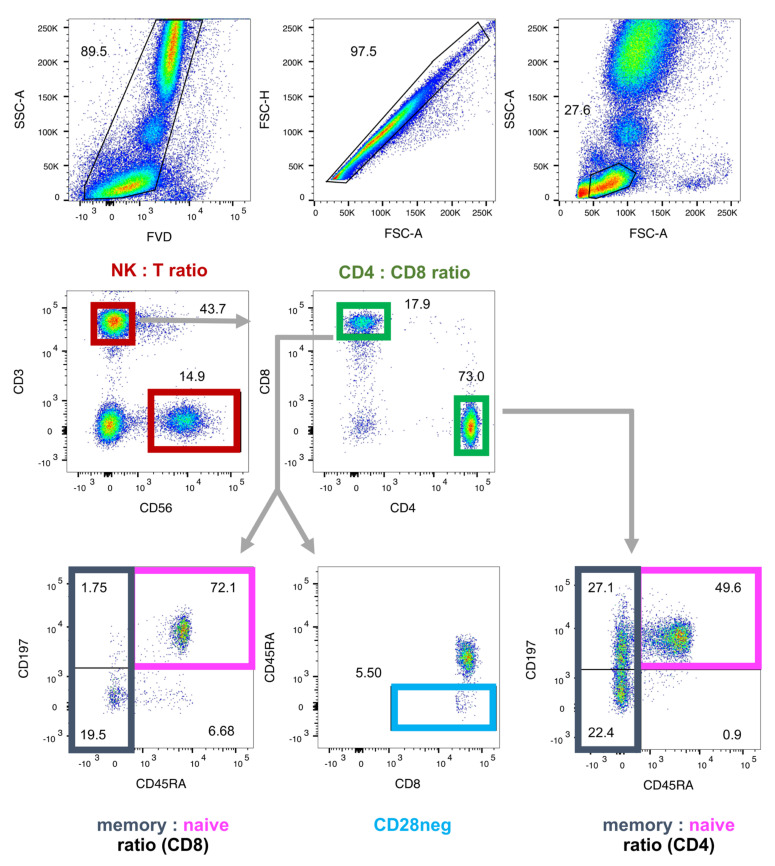
Gating strategy for the derivation of immunosenescence biomarkers by flow cytometry. EDTA blood was stained with a fixable viability dye (FVD) and for the indicated markers, which were calculated from relative cell frequencies: ratios of NK to T cells, CD4pos to CD8pos T cells, and memory to naïve CD4 and CD8 T cells, respectively, and the frequency of CD28neg–CD8pos T cells.

**Table 1 ijms-24-13186-t001:** Materials for PBMC staining, including antigens, antibody clones and coupled fluorochromes, distributors, and antibody dilution used to stain 0.2 × 10^6^ PBMCs (DVS) or 100 µL whole blood (VAC).

Sample	Antigen	Clone	Fluorochrome	Company	Dilution 1/x
DVS	live/dead		zombie Yellow	Biolegend (San Diego, CA, USA)	1000
	CD3	UCHT1	BV510	BD Horizon™ (Franklin Lakes, NJ, USA)	400
	CD56	B159	PE-CF594	BD Pharmingen™ (Franklin Lakes, NJ, USA)	100
	CD4	RPA-T4	APC-H7	BD Pharmingen™	100
	CD8	RPA-T8	FITC	BD Pharmingen™	200
	CD197 (CCR7)	150,503	Alexa Fluor^®^ 647	BD Pharmingen™	50
	CD45RA	HI100	Alexa Fluor^®^ 700	BD Pharmingen™	400
	CD28	CD28.2	PerCP-Cy™ 5.5	BD Pharmingen™	100
VAC	live/dead		Fixable Viability Dye eFluor™ 780	ThermoFisher Scientific (Waltham, MA, USA)	400
	CD3	UCHT1	BV510	BD Horizon™	100
	CD56	B159	PE-Cy™ 5	BD Pharmingen™	50
	CD4	RPA-T4	BV421	BD Horizon™	100
	CD8	RPA-T8	BB515	BD Horizon™	400
	CD197 (CCR7)	3D12	PE	BD Pharmingen™	100
	CD45RA	HI100	Alexa Fluor^®^ 700	BD Pharmingen™	100
	CD28	CD28.2	PerCP-Cy™ 5.5	BD Pharmingen™	100

## Data Availability

The data and analysis scripts for reproducing the results presented in this study are stored in a repository for download at https://doi.org/10.17605/OSF.IO/5AGF4.

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
