# Peer review of "From Immunosenescence to Aging Types—Establishing Reference Intervals for Immune Age Biomarkers by Centile Estimation"

_ijms, 2023, doi:10.3390/ijms241713186_

Round 1
Reviewer 1 Report
The authors present an interesting pilot analysis to better track immunological aging in humans using participants from the DVS. It was interesting that sex was non-significant. This could be further discussed as others are finding sexual dimorphic changes in immunity with age. Either the IMMAX is insensitive to these changes or possibly the sample sizes were too small to capture this. But nonetheless, the data is compelling and the manuscript is worthy of publication.
Author Response
We like to thank this reviewer for the positive feedback concerning the general value of our paper. We further agree with this reviewer, that the non-significant sex effect was an interesting finding, especially since we had shown previously [Ref.20] that females exhibited lower IMMAX values compared to males when adjusting for chronological age. As this result concurs with corresponding data from the original IMM-AGE study [11], we believe that the non-significant sex effect for the follow-up examinations in this study was attributable to the limited sample size (N=53) in this preliminary analysis. Thus, we have added corresponding text to the Discussion section, which now reads as follows:
“Although females had exhibited lower age-related levels of immunosenescence biomarkers, including IMMAX, in previous cross-sectional studies [11,20], we could not demonstrate any significant sex-related longitudinal difference. However, this might be due to the limited sample size, and thus, these preliminary findings will require confirmatory analyses with the completed follow-up examinations.”
Please also refer to the changes in detail as highlighted in the revised manuscript and in the attached rebuttal letter.

Reviewer 2 Report
The authors analyzed the immunosenescence process with an epidemiological approach.
In a study involving a cohort of 1,605 individuals ranging in age from 18 to 97 years, the authors computed the comprehensive IMMmune Age indeX (IMMAX) using flow cytometry-11 analysis to assess various blood cell sub-populations. By utilizing the 50th percentile as a reference point, they converted IMMAX into equivalent years-of-life (EYOL) units and subsequently calculated the immunological age gap, highlighting the disparity between EYOL and chronological age.
However, the paper lacks a broader contextualization of the research and fails to clearly address a pivotal question in applied research: What significance does this study hold for aging research? How can this data contribute to predicting the aging process?
Furthermore, the paper would benefit from including a statement that underscores the significance of the findings for individuals working in the fields of senescence and aging, especially those who may not be familiar with the methodologies employed. This inclusion could broaden the reach of the paper to a wider readership.
In conclusion, it is essential for the authors to revamp the discussion section, making it more accessible and comprehensible to readers who may not be experts in the field. This effort to enhance clarity and context will significantly improve the overall impact of the research.
Author Response
We like to thank this reviewer for the prudent comments and agree with this reviewer’s summary of our study. We appreciate the valuable hints for improving the outreach of our paper to a wide audience, which we have considered in our revised manuscript as detailed below.
We understand the concern related to a broader contextualization of our research. First, in response to the specific questions of this reviewer regarding the significance of our data for aging research, we have added a corresponding paragraph concluding the introduction section as follows
“Although our preliminary findings will require confirmatory analyses with the completed follow-up examinations, the immunosenescence biomarker IMMAX together with the derived metrics EYOL and age gap showed the potential to satisfy the desiderata for clocks representing the rate of immunological aging in longitudinal research [1].”
Following the detailed advice of this reviewer, we have in addition extended the discussion section recapitulating the context and the major objectives of our study at the beginning by introducing the requirements and desiderata for immunological aging clocks, and concluding with a statement on the relevance of our findings for future research related to the aging immune system.
Thus, in order to improve clarity and context, we have accordingly rebuilt the first paragraph of the discussion section, which now recaps the desiderata for immune aging clocks as well as the major objectives of our study, and reads as:
“Aging clock design including immunosenescence biomarkers should preferably involve longitudinal measurements [1]. In addition, it should allow for the categorization of the rate of aging of the individual immune system by defining age-specific reference intervals [24,26]. Of particular interest in this regard will be the comparison of individual clock levels to a defined reference population mapping the biomarker to a years-of-live scale for the easy calculation of the age gap as the difference in years between the biological and chronological age [1]. The immunosenescence biomarker IMMAX [20] will likely comply with the first requirement, because it approximates the IMM-AGE score, constructed from trajectories capturing the individual's immune-aging process [11]. Satisfying the additional desiderata concerning IMMAX as composite immune age metric was the major objective of this study.
Our aims were accomplished by adopting concepts from pediatric growth-chart modelling [27]…”
Furthermore, we have added a concluding paragraph to the Discussion section, including a statement concerning the relevance of our findings for immunological aging research, which reads as:
“In conclusion, we believe that the immune age index IMMAX will have the potential to satisfy the desiderata for immunological aging clocks in longitudinal research [1,24]. In combination with the metrics EYOL and age gap established from our IMMAX data, it will facilitate communicating research findings involving the aging immune system on a familiar years-of-life scale to a broad audience including stakeholders in the scientific community, patient groups and policy makers.”
Please also refer to the changes in detail as highlighted in the revised manuscript and in the attached rebuttal letter.

Round 2
Reviewer 2 Report
Authors addressed my concerns